# Level of shared decision making and associated factors among patients with mental illness in Northwest Ethiopia: Explanatory sequential mixed method study

**Agmas Wassie Abate**[1]*, **Wondimnew Desalegn**[2], **Assefa Agegnehu Teshome**[3], **Aklile Tsega Chekol**[4], **Mastewal Aschale**[4]

1 Department of Psychiatry, Dr. Ambachew Memorial Hospital, Amhara Regional Health Bureau, South Gondar Zone, Ethiopia, 2 Department of Public Health, College of Health Science, Debre Tabor University, Debre Tabor, Ethiopia, 3 Department of Biomedical Science, College of Health Science, Debre Tabor University, Debre Tabor, Ethiopia, 4 Department of Psychiatry, College of Health Science, Hawassa University, Hawassa, Ethiopia

* agmaswasie@gmail.com

**Data Availability Statement:** All relevant data are within the manuscript and its Supporting information file.

## Abstract

### Background

Shared decision-making is a patient–centered and a recovery-oriented mental health system in which consumers are encouraged to actively engage in illness management. Although shared decision-making research in mental health has evolved rapidly in the past two decades, there is a lack of studies examining the level and factors associated with shared decision-making practice in low-income countries like Ethiopia.

### Methods

An institutional-based explanatory sequential mixed method study design was conducted from July 18 to September 18, 2022, at Bahir Dar city specialized hospitals. A systematic random sampling technique was used. The level of shared-decision making was measured by 9-item shared decision-making questionnaire among 423 patients with mental illness. Epicollect5 was used to collect data, which was then exported to the Statistical Package for social science version 25 for analysis. Variables with a P-value < 0.25 were considered candidates for the multivariate logistic regression analysis. The odds ratio with a 95% confidence interval was used to show the strength of the association. An in-depth interview was conducted among ten purposively selected participants.

### Result

Low shared decision-making practice was found to be 49.2% (95% CI 45.9%-55.7%). The Multivariate analysis showed that low perceived compassionate care (AOR = 4.45; 95%CI 2.52–7.89), low social support (AOR = 1.72; 95% CI 1.06–2.80), and no community-based health insurance (AOR = 1.96; 95%CI I.04–3.69) were positively associated with low shared

**Funding:** The author(s) received no specific funding for this work.

**Competing interests:** The authors have declared that no competing interests exist.

decision making. The qualitative result showed that the most common barriers to shared decision-making were a lack of empathy and a shortage of mental health workers.

## Conclusion and recommendation

Almost half of the patients had low shared decision-making practices. This implies that shared decision-making requires high attention as it is essential for patient-centered care.

## Introduction

Shared decision-making (SDM) is a practice by which clients and providers share information and opinions, talk about each other's responsibilities, and ultimately agree on treatments [1, 2]. Shared decision-making research in mental health has evolved rapidly in the past two decades, presenting one of the fastest growth curves in SDM research and practice [3]. There is increasing gratitude that SDM is vital as a decision-making model in clinical practice when more than one preference is medically significant or when patient preferences vary strongly [3–5]. SDM has been shown to improve patient satisfaction and adherence to therapy, and may also reduce undesired care [6, 7].

Shared decision-making promotes a collaborative process for planning care through dialogue among the individual who is affected, caregivers, and clinicians [8]. Three key steps of SDM for clinical practice, namely: choice talk, options talk, and decision talk, where the clinician supports deliberation throughout the process [9].

A study showed that 60% of patients reported that they had experienced shared decision-making during treatment and it was most likely to be associated with affective-cognitive patient outcomes (54%), compared with 37% of behavioral and 25% of health outcomes [10, 11]. Shared decision-making was associated with higher scores on the bond subscale of the Working Alliance Inventory, indicating a higher degree of liking and trust, and better medication adherence [2, 11].

Evidence showed that shared decision-making can be successfully practiced in psychiatry would contribute to an improved inclusion of psychiatric patients in therapeutic decisions and thereby help to implement the basic rights of a group of patients who have not sufficiently benefited from consumer empowerment in other medical fields [12]. Non-psychiatric patients were 18 times more likely to prefer options for their treatment and 2 times more likely to prefer to take medical decisions on their own than psychiatric patients [13]. The primary outcomes of SDM in mental health often differ from outcomes in non-mental health studies (such as SDM use, knowledge, and conflict level), and involve a focus on goals important to the field, such as empowerment, self-determination, and recovery [14–16]. In mental health care, SDM has been recommended for people with mental illness, given that self-determination, choice, and autonomy are core aspects of recovery-oriented care [14].

According to a study done among adult admitted patients at Public Hospitals of West Showa Oromia, Ethiopia, age, marital status and education level were significantly associated with shared decision making [17]. Male patients, patients with diagnoses involving psychotic symptoms, patients with longer treatment durations, and involuntary treated patients experienced less SDM [18].

Assessment of shared decision-making in the outpatient psychiatry department has great importance for improving quality health service delivery by focusing on identifying factors that affect shared decision-making with mental health services. It will help to increase the

treatment-seeking behavior of the patient. In addition, there was no similar research completed in Ethiopia. It helps as a source of data and information for stakeholders and researchers who work in the area of mental health services and those who will conduct further studies on the issue under inquiry.

# Methods and material

## Study area and period

This study was conducted at Tibebe Ghion specialized hospital (TGSH) and Felege Hiwot Comprehensive Specialized Hospital (FHCSH) in Bahir Dar city, Ethiopia. TGSH hospital gives mental health services with four outpatient departments, one emergency room, and two outpatients with a total bed of 13. It also gives psychotherapy services by a clinical counselor. The community is served by two psychiatrists, seven mental health specialists, one clinical counselor, and six psychiatric professionals. The estimated number of annual outpatient clients was 4864 (405 per month based on the monthly report of the psychiatry unit). FHCSH also provides mental health services with inpatients (17 beds) and four outpatient departments. It serves a total of 19,200 clients annually (1600 per month). The patient was served by four mental health specialists and six psychiatry professionals. Generally, the monthly patient flow for both hospitals in psychiatry outpatient was 2005 patients. The study was conducted from July 18/2022 to September 18, 2022.

## Study design and population

An institutional-based explanatory sequential mixed study design was used. All patients who visited the outpatient department (OPD) for mental health services at TGSH and FHCSH who were available at the time of data collection and whose age was 18 years or older were included. Those patients who were seriously ill were excluded. Four patients were excluded due to they did not give response due to their active psychopathology. They were referred to the Emergency Psychiatry ward.

## Study variables

Shared decision making (Good/ Poor) was the outcome variable. The independent variables were socio-demographic (age, sex, residence, occupation, marital status, monthly income, educational level and religion), clinical related (current diagnosis of mental illness, duration of the illness, number of episode and type of visit) and psychosocial related factors (social support, perceived stigma, perceived compassionate care and patients anticipated stigma from heath workers).

## Sample size determination

Since no studies have been done in Ethiopia on shared decision-making, we have used a proportion of 50%. By using a single population proportion formula with a 95% confidence interval and a 5% margin of error, the sample size was calculated by adding a 10% non-response rate, the final sample size was 423. To get representative data in both hospitals, proportional allocation of the sample was calculated based on patient flow per month as 405/1600, and the proportion was 1:4. The final sample size distribution was 106 for TGSH and the rest 317 for FHCSH (Fig 1).

For qualitative research, 10 participants were selected purposively for the in-depth interview after analysis of the quantitative data, and until information saturation was obtained.

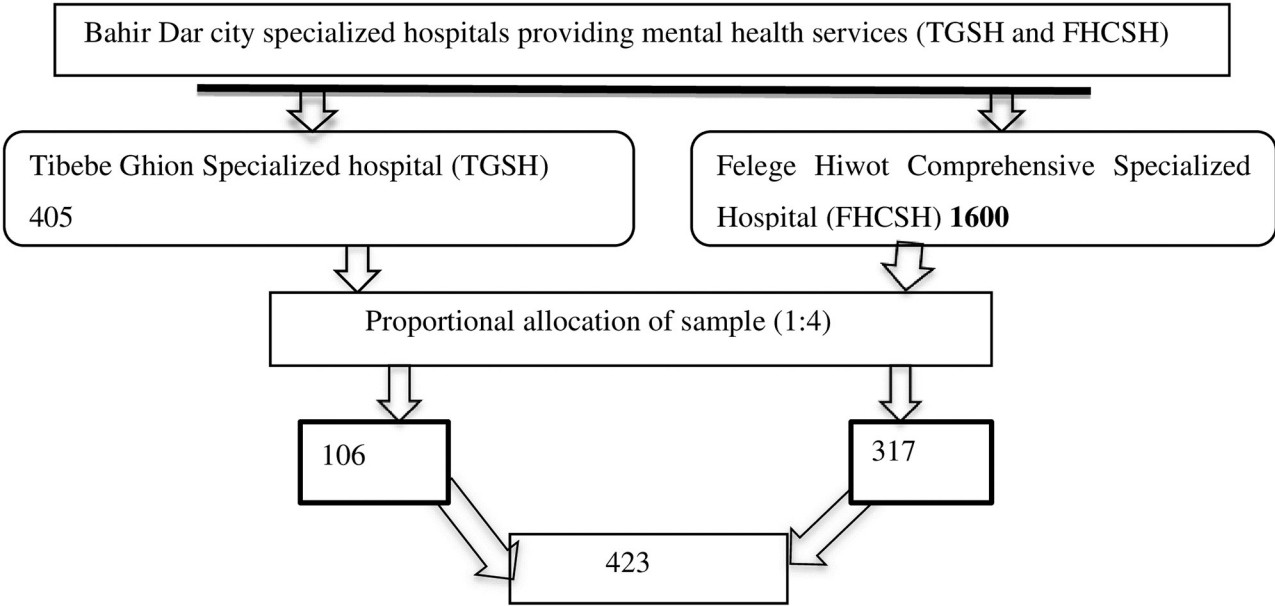

**Fig 1. Schematic presentation of proportional allocation of sample among patients with mental illness North West, Ethiopia, 2022.**

## Operational definitions

**Shared decision making.** This is measured by the core instrument, which consists of nine items, which can be rated a six-point scale from "completely disagree" (0) to "completely agree" (5). Summing up all items leads to a raw total score between 0 and 45, this was transformed to 0 to 100 by multiplying it by 20/9. The mean was computed and used as a cut point: High > 54.48 and Low <54.48 [19].

**Perceived compassionate care.** It is critical to high-quality health care and was measured by the Schwartz Center Compassionate Care Scale® (SCCCS), which assesses the level of perceived compassionate health care and categorized as high; a score greater than the mean (76.74) and low; a score less than the mean (76.74) [20–22].

**Social support.** Measured by OSLO 3 social support which is categorized into poor social support (3–8), moderate social support (9–11) and strong social support (12–14) [23].

**Perceived stigma.** Measured by the Stigma Scale for Receiving Psychological Help, a five-item scale with an internal Cronbach's alpha of 0.74, was created by summing the scores (0–3) for each item, ranging from 0, the lowest perceived stigma, to 15, the highest perceived stigma; then the cut point is 6.86. The score above 6.86 was considered as having high perceived stigma and the score below 6.68 considered as having a low perceived stigma [24].

**Patient anticipated stigma from health workers.** A four-item survey with a Likert scale ranging from 1 (very unlikely) to 5 (very likely) is used to assess patient anticipated stigma to health care workers. Summing up all items leads to a total score of 4 to 20. The mean score above 10.89 was regarded as high patient anticipated stigma to health workers, while the score below 10.89 was regarded as low patient anticipated stigma to health workers [25].

## Sampling technique and procedure

A systematic random sampling technique for quantitative studies was applied to recruit the study samples. The two hospitals were included in the study. The proportional allocation of

samples was done based on patient flow per month in each hospital, and the proportion was 1:4 (Fig 1).

For the qualitative study, a purposive sampling technique was employed, and 10 participants (6 from patients and 4 from psychiatry professionals) were selected based on their repetitive follow-up after the analysis of quantitative data.

### Data collection methods and tools

The questionnaire consists of semi-structured interviewer-administered questions. Shared decision-making was measured by 9-item Shared Decision-Making Questionnaire (SDM-Q-9) with internal consistency and yielded a Cronbach's value of = 0.938 [19]. The tool consists of nine items, which can be rated on a six-point scale from "completely disagree" (0) to "completely agree" (5). Summing up all items leads to a raw total score between 0 and 45, then transformed to 0 to 100 by multiplying 20/9. The mean was computed and used as a cut point: agree > 54.48 and disagree <54.48 [19]. Socio-demographic, clinical, and service-related variables were included in the questionnaire. Four BSc psychiatric nurses used Epicollect5 software to collect data, which was supervised by two psychiatry unit coordinators, one from each of the two hospitals.

### Data collection methods and tools for the qualitative study

After the analysis of quantitative data, unstructured questions (interview guide) were prepared by the principal investigator by reviewing different articles. Then in-depth interviews were conducted by maintaining their privacy and using a mobile recorder to record the interview process.

### Data quality control

The semi-structured interviewer-administered questionnaire was prepared in English first and then changed to the local language (Amharic) for a better understanding of the data collectors and the respondents, and then it was changed back to English again to check its consistency. The one-day training was given for data collectors on the objective of the study, data collection tools and procedures, how to approach potential respondents, and how to maintain confidentiality. Pretesting on 5% of the sample size (21 participants) was done prior to a week of the actual data collection period at Addis Alem General Hospital. The final data collection tool was refined based on the findings from the pretest. The collected data was carefully checked for completeness and consistency on a day-to-day basis.

Recorded materials were listened to repeatedly for familiarization, and transcribed data was read extensively before considering the thematic areas of the study for better exploration. Every point recorded in the notebook was counterchecked with the phone recorded for consistencies.

### Data processing and analysis

Epicollect5 was used to collect data, which was then directly exported to Statistical Package for social science (SPSS) version 25 for further analysis. Descriptive statistics such as mean, standard deviation, proportions, frequency, and percentage were used to describe the outcome and independent variables. Variables that were shown to have an association with the dependent variable in the bivariate analyses at P<0.25 were candidates for the multivariable logistic regression model. The adjusted odds ratio (AOR) with 95% CI and a p-value less than 0.05 was used to declare the strengths and the factors significantly associated with the outcome variable

respectively. Model fitness was checked, which was (Hosmer and Lemeshow Test) = 0.10, no multicolinearity (Tolerance>0.1 and Variance inflation factor (VIF) <3).

For the qualitative part, after repeatedly listing to the audio recordings, data was transcribed and translated to English, and then it was read extensively. Thematic analyses using QDA miner lite software were done. Finally, the findings were triangulated with the quantitative one.

### Ethics approval and informed consent

Ethical clearance was obtained from institutional review board (IRB) of Bahir Dar University College of Medicine and Health Science then obtain permission letter, it was given to TGSH and FHCSH. The protocol number from IRB decision was 486/2022. The study subjects were approached individually and give information regarding the purpose of the study and then verbal consent was obtained for the study. Then the importance and confidentiality of the information gather was explained to each of the competent participant before the start of interview. Participants were also informed that they were never get any benefit because of participation in the study and no harm on them if they were not agreeing to participate or withdraw from participation during the data collection.

## Result

### Socio-demographic characteristics of participants

A total of 419 patients with mental illness were involved in this study, with a response rate of 99%. Two hundred thirty (50.8%) participants were females and one hundred fifty six (37.2%) of the participants were aged 26–35 years, with a mean age of 33.02 (±SD of 11.2). The majority of the participants (98.6%) were Amhara in ethnicity, 50.8% of the participants were lived in urban areas, and 26.0% were unable to read and write. Two hundred ninety six (70.6%) were Orthodox Christians, and 30.1% were farmers. Moreover, 47.5% and 28.2% of the participants were married and had a monthly income <2000 birr (with SD ± 3529 birrs) respectively (Table 1).

### Clinical related characteristics

The majority of the participants (89.3%) were in follow-up and 37.5% of the participants attended their follow-up with a diagnosis of schizophrenia. 90.2% of the participants had 1–2 episodes and 59.7% had duration of illness less than 24 months before the initiation of their treatment, with the median duration of illness of 24.00 months (Table 2).

### Social related characteristics

One hundred ninety two (45.8%) participants had strong social support, and two hundred twenty eight (54.4%) of the participants had high perceived and anticipated stigma toward health workers (Table 2).

### Service-related characteristics

Three hundred thirty nine (80.9%) of participants had family involvement during their treatment, 57.3% of the participants came from a distance of more than 30 kilometers and 51.6% had community-based health insurance (Table 2).

**Table 1. Socio-demographic characteristics of respondent's care among patients with mental illness at TGSH and FHCSH, North West, Ethiopia, 2022 (n = 419).**

| Variables | Category | Frequency | Percent (%) |
|---|---|---|---|
| Sex | Male | 206 | 49.2 |
| | Female | 213 | 50.8 |
| Age | 18–25 | 133 | 31.7 |
| | 26–35 | 156 | 37.3 |
| | 36–25 | 89 | 21.2 |
| | ≥46 | 41 | 9.8 |
| Residence | Urban | 213 | 50.8 |
| | Rural | 206 | 49.2 |
| Ethnicity | Amhara | 413 | 98.6 |
| | Other* | 6 | 1.4 |
| Religion | Orthodox | 296 | 70.6 |
| | Muslim | 82 | 19.6 |
| | Protestant | 41 | 9.8 |
| Marital status | Married | 199 | 47.5 |
| | Single | 157 | 37.5 |
| | Divorced | 53 | 12.6 |
| | Other** | 10 | 2.4 |
| Educational status | Cannot read and write | 109 | 26 |
| | Primary education | 109 | 26 |
| | Secondary education | 109 | 26 |
| | Diploma and above | 92 | 22 |
| Occupation | Farmer | 130 | 31.1 |
| | Private | 128 | 30.5 |
| | Gov't employed | 39 | 9.3 |
| | Student | 64 | 15.3 |
| | Daily laborer | 18 | 4.3 |
| | Other*** | 40 | 9.5 |
| Monthly income in the household(ETB) | <2000 | 116 | 27.7 |
| | 2001–4000 | 118 | 28.1 |
| | 4001–6000 | 113 | 27.0 |
| | >6001 | 172 | 17.2 |

*Oromo

**separated, Widowed

***Housewife, Jobless, ETB = Ethiopian Birr

## Level of shared decision making

Overall, two hundred six (49.2%) (95% CI = 45.9%-55.7%) of participant's had low level of shared decision-making involvement during treatment at TGSH and FHCSH (Fig 2). From two hundred six patients with mental illness, 39.3%, 27.2%, 18.9% and 7.8% had poor decision making involvement during their psychiatric outpatient follow-up among patients diagnosed with Schizophrenia, Major Depressive Disorder (MDD), Bipolar disorder (BPI disorder), and Generalized Anxiety Disorder (GAD) respectively (Fig 2).

## Factors associated with shared decision making

The multivariable analysis showed that poor perceived compassionate care, no community based health insurance, and poor social support were significantly associated with low shared

**Table 2. Clinical, social, and service related characteristics of respondent's care among patients with mental illness at TGSH and FHCSH, North West, Ethiopia, 2022 (n = 419).**

| Variables | Category | Frequency | Percent (%) |
|---|---|---|---|
| Diagnosis | BPI disorder | 73 | 17.4 |
|  | Schizophrenia | 157 | 37.5 |
|  | MDD | 124 | 29.6 |
|  | GAD | 29 | 6.9 |
|  | Other* | 36 | 8.6 |
| Type of service | Follow-up | 374 | 89.3 |
|  | New | 45 | 10.7 |
| Duration of illness (Months) | <24 | 250 | 59.7 |
|  | 24–36 | 90 | 21.5 |
|  | ≥36 | 79 | 18.9 |
| Episode | 1–2 | 378 | 90.2 |
|  | >2 | 41 | 9.8 |
| Family involvement | Yes | 339 | 80.9 |
|  | No | 80 | 19.1 |
| CBHI | Yes | 216 | 51.6 |
|  | No | 203 | 48.4 |
| Distance from the Hospital | <10 | 102 | 24.3 |
|  | 10–30 | 92 | 22.0 |
|  | >30 | 225 | 53.7 |
| Social support | Strong | 192 | 45.8 |
|  | Moderate | 127 | 30.3 |
|  | Poor | 100 | 23.9 |
| Perceived stigma | High | 228 | 54.4 |
|  | Low | 191 | 45.6 |
| Perceived compassionate care | Good | 199 | 47.5 |
|  | Poor | 220 | 52.5 |
| Patient anticipated stigma towards health workers | High | 228 | 54.4 |
|  | Low | 191 | 45.6 |

(*Panic disorder, Schizophreniform, Dementia, Epilepsy, Dysthymia, Posttraumatic stress disorder) BPI: Bipolar I disorder MDD: Major depressive disorder GAD: Generalized anxiety disorder CBHI: community-based health insurance)

decision-making. The odds of low shared decision-making among patients with mental illness who had low perceived compassionate care was 4.45 times (AOR = 4.45; 95%CI 2.52–7.87) more likely than among patients who had high perceived compassionate care. The odds of low shared decision-making among patients with mental illness with poor social support was 1.7 times (AOR = 1.72; 95% CI 1.06–2.80) higher than among patients with strong social support. The odds of low shared decision-making among patients with mental illness with no community based health insurance was 1.96 times (AOR = 1.96; 95% CI 1.04–3.69) higher than those who had community-based health insurance (Table 3).

## Qualitative findings

On the facilitators and barriers to the shared decision-making; in-depth interviews were conducted with the participants. In addition, to the interview guide, the preliminary quantitative results were used to frame the discussion. Throughout the analysis, three themes have come

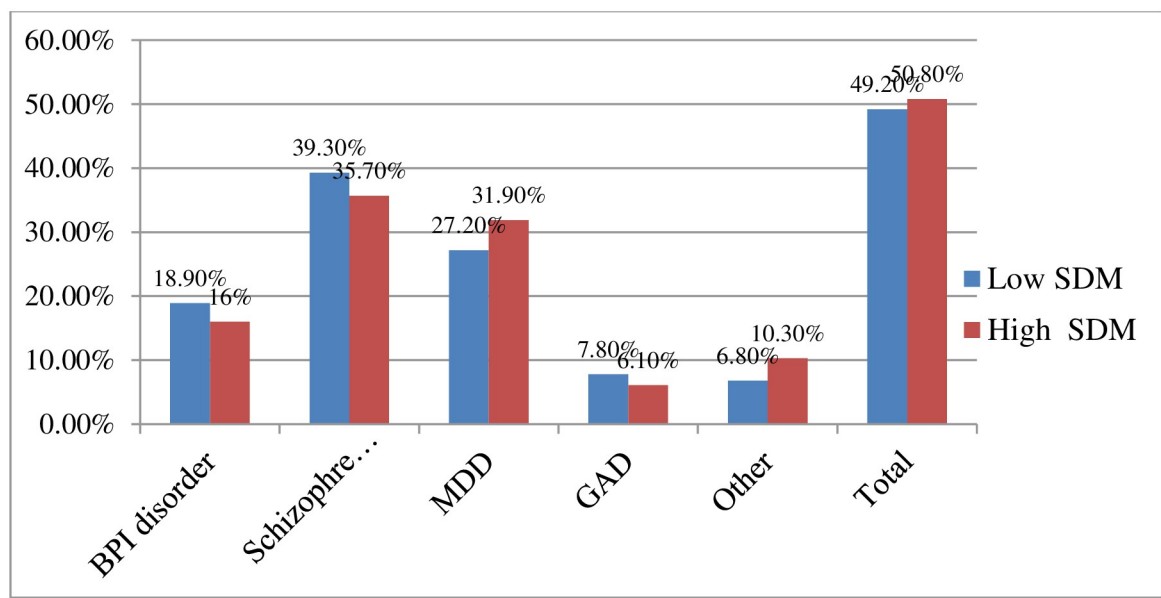

**Fig 2. Schematic presentation of distribution of shared decision making involvement by mental disorders, North West, Ethiopia, 2022.**

into focus. These recurring themes include service quality-related, psychosocial, and human resource-related concerns.

**Service quality-related factors.** When the participants were asked about shared decision-making practice at the hospital, they stated that good compassionate care was important for health service quality which is the facilitator for shared decision-making practice.

*"... I have been in follow-up for more than 10 years, but I have been open to professionals and they treat everyone equally. Learning means faith, so at this stage; there is no discrimination based on religion or others. Compassion, in my opinion, means looking at everyone equally and reaching on the common decision for my treatment"* (**IDI5**-25-year-old male). *"... Mental health professionals treat me with a polite and good approach. In my follow-up, the psychiatrists who work with me understand my mental illness and write medical certificate for the office where I work to do the job that is appropriate for my capacity"* (**IDI2**-27-years-old female). *"...I felt very happy that the professionals compassionately treated me because when I was treated at another health facility, they asked me from a distance: no expert to help me in a good approach and without involvement in the overall treatment, but in this hospital, they received me well and treated me with other patients equally"* (**IDI3**-32-year-old female).

To practice good shared decision-making practice lack of empathy is described as the main barrier for it.

*"... One scenario was that I accidentally stopped taking the medicine when my mother got sick, and then my illness came back. When I told the mental health worker that I had stopped taking the medication and that the illness had returned, he told me that if he saw me in illness now, he would give me the medication... When I sat down, he left me. At first, the gap could have been on me. But as a professional, he should have understood me and given me proper advice so that I don't repeat the gaps. At the center, when the professional left me, I was angry, and it hurt me there. I don't know how they treated me after those occasions"* (**IDI4**-25 year-old female). *"... Empathy means treating patients with respect, and the care provided by hospital psychiatrists and mental health professionals is often rough. Some professionals are empathetic,*

**Table 3. Bi-variable and multivariable binary logistic regression analysis showing the association between shared decision-making involvement and associated factors among patients with mental illness at TGSH and FHCSH (n = 419).**

| Variables | Category | Shared decision making (level) | | COR(95% CI) | AOR(95% CI) |
|---|---|---|---|---|---|
| | | Low | High | | |
| Perceived compassionate care | Low | 150 | 70 | 5.47(3.60,8.32) | 4.45(2.52,7.87)* |
| | High | 56 | 143 | 1.00 | 1.00 |
| Residence | Rural | 112 | 94 | 1.51(1.51,1.03) | 1.76(0.99,3.12) |
| | Urban | 94 | 119 | 1.00 | 1.00 |
| Monthly income(ETB) | <2000 | 56 | 60 | 0.91(0.50,1.64) | 0.84(0.38,1.90) |
| | 2001–4000 | 61 | 57 | 0.79(0.44,1.42) | 0.82(0.36,1.84) |
| | 4001–6000 | 56 | 57 | 0.86(0.48,1.56) | 0.95(0.45,2.02) |
| | >6001 | 33 | 39 | 1.00 | 1.00 |
| Duration of illness (months) | <24 | 118 | 32 | 1.21(0.73,2.00) | 0.65(0.34, 1.23) |
| | 24–36 | 47 | 43 | 0.99(0.54,1.81) | 0.64(0.31,1.33) |
| | >36 | 41 | 38 | 1.00 | 1.00 |
| Family involvement | Yes | 157 | 182 | 1.00 | 1.00 |
| | No | 49 | 31 | 0.55(0.33,0.90) | 1.20(0.65,2.21) |
| CBHI | Yes | 93 | 123 | 1.00 | 1.00 |
| | No | 113 | 90 | 1.44(0.98,2.11) | 1.72(1.06,2.80)** |
| Social support | Poor | 66 | 34 | 3.38(2.04,5.62) | 1.96(1.04,3.69)** |
| | Moderate | 70 | 57 | 1.58(0.92,2.72) | 1.63(0.88,3.01) |
| | Strong | 70 | 122 | 1.00 | 1.00 |
| Perceived stigma | High | 133 | 95 | 2.26(1.53,83.35) | 0.86(0.46,1.61) |
| | Low | 73 | 118 | 1.00 | 1.00 |
| Sex | Male | 96 | 110 | 1.22(0.83,1.80) | 1.07(0.67,1.72) |
| | Female | 110 | 103 | | |
| Patient anticipated stigma to health workers | High | 140 | 88 | 3.01(2.02,4.50) | 1.68(0.92,3.08) |
| | Low | 66 | 125 | 1.00 | 1.00 |
| Episode | 1–2 | 183 | 196 | 1.36(0.71,2.61) | 1.34(0.60,2.99) |
| | >2 | 23 | 18 | 1.00 | 1.00 |
| Distance(km) | <10 | 55 | 47 | 0.82(0.31,1.37) | 0.42(0.21,0.83) |
| | 10–30 | 41 | 51 | 1.19(0.73,1.94) | 0.93(0.49,1.76) |
| | >30 | 110 | 115 | 1.00 | 1.00 |

*p<0.001

**p<0.05.

CBHI: community-based health insurance, ETB: Ethiopian Birr

and some are not. The main reason for this problem was the negligence from mental health professionals" (**IDI1**-18 year old male).

**Psychosocial related factors.** Strong social support was also explored by IDI as having a relationship with good shared decision-making practices and as it increases the client and health professional bonding. "... *I am deeply grateful to my family for enabling me to start psychiatric treatment. I never thought that other people and professionals would ostracize me for having a mental illness. Because of this, taking my medication without stopping has helped me recover from my illness faster. Therefore, it fosters shared decision making practices, which can help patients recover faster and measure the patient-centered quality of hospital care"* (**IDI6**-25-years-old male). Others noted that there should be a mechanism to strengthen social support

in hospitals. *". . .One scenario* is that *I am going to stop taking this medicine because the price of the medicine is high. I did not come up with enough money; no one supported me. As a result, it creates a barrier between me and health professional's relationship"* (IDI2-27-year-old female).

Low perceived stigma was described by the participant as it increases the treatment-seeking behavior of those who have been linked with good shared decision-making practice. *". . . One scenario was that if we go to a rural area when they find out that someone has a mental health problem, they will point it out to me on the road. But I don't mind when people point it out to me; I would not be too sentimental and I am confidential to continue my follow-up as long as the health professionals care is by understanding my suffering, but according to some patients, they do not dare go to treatment for mental illness because they are afraid that someone else may know about their illness"* (**IDI5**-25 year old male). *". . . Friends greet me and take care of me "***(IDI6-23 year's old male)**.

According to IDI, low patient anticipated stigma is also described as it increases the treatment-seeking behavior and is related to good shared decision-making. *". . . I never thought that other people and professionals would ostracize me for having a mental illness rather the professional's make me to feel happy with my follow-up"* (**IDI6**-23- years-old male). *". . . Practitioners treat me by understanding my suffering rather than blaming me; it helps to cure the disease quickly. Professionals are kind enough to provide me with a way to protect my mental illness. In my opinion, medical service is provided on the basis of equality* (**IDI5**-25-years old male).

**Human resource-related factors.** Lack of training/supervision and a shortage of mental health workers were elaborated as the main barriers to shared decision-making practices.

*". . . "Close support and monitoring are needed to enable them to receive good care and achieve good outcomes*" (**IDI2**-27 years old). . . . *"In my opinion, they also need training and evaluation of health professionals regarding shared decision making"* (**IDI4-25 year's old female**).

Patient awareness about mental illnesses and the services provided by the participant was described as an essential approach to distinguishing the patient's own rights and responsibilities during their hospital stay, which helps to foster shared decision-making.

*". . .To strengthen shared decision-making in mental health care, patients should first keep their appointments and come to the hospital. Second, patients need to explain their illness to professionals without hiding it, just as it is said that there is no medicine for those who hide their illness. I think it would be good for them to openly explain their illness without hiding it. "(IDI8-24 year old female) Mental health workers were devoting their time to giving us advice; we should listen to them and take our medicine properly." "Mental health education should also be strengthened in every religious institution " (IDI5-25 year old male)". . . . It is good if patients are advised not only financially but also psychologically so that they can be treated on time without being isolated by the community"* (IDI9-**29-year-old male**).

A shortage of mental health workers was noted by the participants, and to solve this problem, additional mental health professionals were needed to solve work overload and foster shared decision-making.

"*. . .* In *order for patients with mental illness to receive good services, it is necessary to have a sufficient number of professionals. The reason is that as the number of patients increases, it is necessary to have enough professionals to provide efficient services with good shared decision making involvement"* (**IDI3**-32-year-old female). *As a result, service speed will increase to allow for adequate time to consult with health professionals* (**IDI2**-27-year-old female).

## Discussion

This study revealed that the low shared decision-making practice at TGSH and FHCSH was 49.2% (95% CI 45.9%-55.7%). This implies that shared decision-making needs high attention

as it is essential for patient-centered care [26]. The finding of this study also showed that low perceived compassionate care, poor social support and no community-based health insurance were associated with low shared decision-making practice.

The level of shared decision making practice in this study was low (49.2%). This was lower than the study done in Public Hospitals of West Showa Oromia, Ethiopia (64.8%) [17] and in Australia with 60% reporting experiencing SDM [11]. This may be due to despite the flourishing development of SDM interventions [27], SDM implementation for patients with serious mental illness has been relatively less successful than for other groups. This disparity has been attributed to a variety of barriers [28], including clinicians' paternalistic approach to the care of patients with serious mental illness and stigma-related beliefs that SDM is inappropriate for such patients. Unlike other chronic non-psychiatric conditions, the characteristics of serious mental illness may pose a major barrier to SDM implementation. For example, psychotic symptoms and cognitive deficits associated with schizophrenia can affect decision-making capacity and serve as barriers to SDM, whereas an elevated blood-glucose level associated with diabetes does not stop patients from being involved in SDM [14, 28].

This result was higher than a study finding of patient attitude in SDM in outpatient psychiatry services where 30% of patient did not experience shared decision-making involvement [11]. The variation may be explained by the settings, the study population, and tool differences used in the studies. This study uses the standardized tool which was the 9-item shared decision-making questionnaires but the previous study uses Mini-Mental State Examination in 109 participants.

This study showed that patients with mental illness with the low level of perceived compassionate care were 4.45 times (AOR = 4.45, 95% CI 2.52–7.89) more likely to get low shared decision making than those patients with the high level of perceived compassionate care. This finding was supported by the qualitative study findings. As IDI1 stated *". . . Compassion means treating patients with respect, and the care provided by hospital psychiatrists and mental health professionals is often rough; not customer-centered. Some professionals are empathetic, and some are not. They were not involving us during the treatment process. The main reason for this problem was the negligence of mental health professionals"* (18 year old male). This finding might be psychiatrists, as well as other medical providers, score low on scales of patient involvement in decision-making, perhaps in part because traditional genetic counseling has been based on autonomous choice models and negative internalized self- stigma which affects perceived compassionate care, a common experience that leads patients to underestimate their own competency for decision-making and autonomy [5, 29, 30].

This study also showed that patients with poor social support were 1.72 times (AOR = 1.72, 95% CI 1.06–2.8) more likely to receive low shared decision-making than those patients with strong social support. This study was supported by the qualitative finding as the participant (IDI2) stated that ". . .One scenario is that *I am going to stop taking this medicine because the price of the medicine is high. I did not come up with enough money; no one supported me. As a result, it creates a barrier between me and the health professional's relationship and it affects shared decision involvement between me and the mental health professional"*. This could be due to strong social support being highly associated with good medication adherence, improved quality of life, increased mental health-seeking behavior, improved treatment outcomes, and increased health professional-to-patient bonding [31] Emotional support includes shared risk-taking and intimacy as well as the provision of caring and empathy [3, 32]. A greater barrier, though, may be clinicians' fear of liability and legal exposure. Clinicians may resist SDM with patients with serious mental illness because they fear being held liable for any potentially negative outcome that might result from SDM, such as symptom exacerbation, hospitalization, or death. Burnout, patient load, and limited appointment time also contribute to clinicians'

reluctance to engage in SDM [28]. This study also indicates that patients who had no community based health insurance were 1.96 times (AOR = 1.96, 95% CI 1.04–63.69) more likely to receive low shared decision-making than those who had community-based health insurance. This finding was explained by the qualitative findings as community-based health insurance helps the patient to choose the preferred treatment without financial constraints. As IDI7 explains, *". . .because of I have had community-based health insurance, I decided together with the experts to accept the medicine given by the expert, and I feel happy that I decided on every aspect with them* (**IDI7)**. This finding may be due to the utilization of health services among insured households with community-based health insurance being higher and is an emerging strategy for providing financial protection against health-related poverty and fostering shared decision making [33, 34].

## Strengths and limitations of the study

The response rate in this study was high, which helped to reduce the probability of non-response. To minimize bias, we used a standardized and pre-tested questionnaire. The study's findings might be prone to a response bias due to the patients self-report, and they do not provide an objective measure of shared decision-making. The questionnaire has some sensitive issues, which may suffer from social desirability. Patients who had admitted in inpatient and Emergency psychiatry ward were not included in this study.

## Conclusion

Two hundred six (49.2%) patients were found to receive low shared decision-making at Tibebe Ghion specialized and Felege Hiwot comprehensive specialized hospitals. This implies that shared decision-making needs high attention as it is essential for patient-centered care. Low perceived compassionate care, poor social support and no community-based health insurance were associated with low shared decision-making practice. Based on the findings of this study, the concerned organization should create regular follow-up and supervision for the implementation of shared decision making in each health service area.

## Supporting information

**S1 Checklist. STROBE statement—Checklist of items that should be included in reports of observational studies.**
(DOCX)

**S1 File.**
(SAV)

## Acknowledgments

We are grateful to the data collectors for their admirable endeavors. Also, our appreciation goes to the study participants who willingly contributed to this study by responding to the questionnaires.

## Author Contributions

**Conceptualization:** Agmas Wassie Abate.

**Data curation:** Agmas Wassie Abate, Wondimnew Desalegn, Assefa Agegnehu Teshome, Mastewal Aschale.

**Formal analysis:** Agmas Wassie Abate, Wondimnew Desalegn, Assefa Agegnehu Teshome, Aklile Tsega Chekol, Mastewal Aschale.

**Methodology:** Agmas Wassie Abate, Wondimnew Desalegn, Aklile Tsega Chekol, Mastewal Aschale.

**Resources:** Assefa Agegnehu Teshome.

**Software:** Wondimnew Desalegn, Assefa Agegnehu Teshome, Aklile Tsega Chekol.

**Validation:** Wondimnew Desalegn, Assefa Agegnehu Teshome.

**Writing – original draft:** Agmas Wassie Abate, Aklile Tsega Chekol, Mastewal Aschale.

**Writing – review & editing:** Agmas Wassie Abate.

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
