## [Decision Letter · Decision Letter 0]

27 Feb 2023

PONE-D-22-33947Level of shared decision making and Associated Factors among Patients with Mental Illness in Northwest Ethiopia: Explanatory sequential mixed method studyPLOS ONE

Dear Dr. Abate,

Thank you for submitting your manuscript to PLOS ONE. After careful consideration, we feel that it has merit but does not fully meet PLOS ONE’s publication criteria as it currently stands. Therefore, we invite you to submit a revised version of the manuscript that addresses the points raised during the review process.

We look forward to receiving your revised manuscript.

Kind regards,

Mesfin Gebrehiwot Damtew

Academic Editor

PLOS ONE

Journal Requirements:

4. Please include a caption for figure 1.

5. Please ensure that you refer to Figure 1 in your text as, if accepted, production will need this reference to link the reader to the figure.

6. We note you have included a table to which you do not refer in the text of your manuscript. Please ensure that you refer to Table 3 in your text; if accepted, production will need this reference to link the reader to the Table.

Reviewers' comments:

Reviewer's Responses to Questions

**Comments to the Author**

1. Is the manuscript technically sound, and do the data support the conclusions?

Reviewer #1: Yes

Reviewer #2: Yes

2. Has the statistical analysis been performed appropriately and rigorously? 

Reviewer #1: I Don't Know

Reviewer #2: Yes

3. Have the authors made all data underlying the findings in their manuscript fully available?

Reviewer #1: Yes

Reviewer #2: Yes

4. Is the manuscript presented in an intelligible fashion and written in standard English?

Reviewer #1: No

Reviewer #2: No

5. Review Comments to the Author

Reviewer #1: The authors are to be congratulated for taking on this important subject. Shared decision-making between psychiatric patients and mental health workers is as important in Ethiopia as it is in England or America. The major impediment to achieving shared decision-making everywhere is the constraint of time; the less time per patient, the more difficult it is to achieve. Shared decision-making assumes that patients believe that there is something wrong and therefore need for decisions regarding treatment. If patients have an impaired awareness of their illness – anosognosia - shared decision-making is impossible to achieve other than agreeing to do nothing.

The main shortcoming of this is that it does not deal with this issue sufficiently. It simply says that “those patients who were seriously ill and had no insight were excluded.” How many were excluded? What percentage of the total sample? Various levels of impaired insight occur most commonly in psychosis. Therefore if a mental health worker perceives or believes that a given patient has impaired insight, the mental health worker is less likely to attempt to achieve shared decision-making. If this is true that I would expect shared decision-making to be lower for patients with a diagnosis of schizophrenia, for example, and for those with a diagnosis of depression or anxiety disorders, for example. This could be easily tested with your existing data.

Other problems are minor. The paper could be improved grammatically by using a native English speaker. And there are a few typos such as ages listed in table 1 and the lack of publication information for reference number 16.

Reviewer #2: Comments to the Author

The following comments and questions are forwarded to the authors based on the assessment.

Introduction

This is an interesting paper on the Level of shared decision making and Associated Factors among Patients with Mental Illness in Northwest Ethiopia: Explanatory sequential mixed method study.

In this section fact and figure is not in logical order. The introduction is not clear enough. In this sense, you need to rewrite your introduction. I also think that there needs to be more clarity about the justification of the study and why you chose to use explanatory sequential mixed method. In general, the authors should revise their manuscript in a scientific way with correct language usage and resubmit their manuscript. Therefore, you should rewrite your introduction by focus on your topic taking into consideration the following points that to be raised.

1. What is it? The problem

2. Magnitude and distribution?

3. Possible factors?

4. Severity and consequences?

5. Gap statement?

Keywords: Minor edition, make the first letters of all keywords uppercase.

Methods

Study design?

Sample size determination

No need to present sample size formula rather narrate how the sample size calculated. Remove the formula and definition of symbols or alphabet used in the sample size calculation.

Sampling technique and procedure

Figure 1 is the not figure in the scientific writing. Please, recreate figure 1 in attractive way by arranging the alignment of arrow and rectangles used.

Study variable

Where is study variables?

Operational definition

If you present operational definition for the outcome variable “Shared decision making(level)” (What it mean low or high Shared decision making) in methodology part is more appropriate for scientific paper, and the way you measured the outcome variable (scoring, including categorization and how composite index or total score formed) under data collection with the variables measured. Thus, I recommend to present the operational definition of Shared decision making level, and provide the method of measuring the Shared decision making. Most importantly provide your justification why mean score was used as cut-off point with reference. Who recommend the cut-off point?

Results

Change Quantitative finding to sociodemographic characteristic of the respondents.

If you are presenting the exact number remove the words “about, nearly, majority…”, Because you are writing the exact number and the reader can evaluate by themselves.

Results presentation is not presented in the way readers can understand. Just a list of sentences without creativity. The authors should read previously published similar article and represent the results part clearly.

Discussion

The discussion part is not presented in the way guide the readers and clear way. In the first part of the discussion part, the authors should summarize the key findings which they are going to discuss. However, the authors just stated their discussion by comparing their finding with the previous findings. Even the way they are comparing with previous finding is not clear. Thus, the authors should read the previous study or any scientific writing books and try to summarize the key findings in the first paragraph of the discussion and then start results interpretation, and compare and contrast with previous studies. I strongly recommend for the authors should follow the previous study and try to present the discussion. In fact not only the discussion whole manuscript should be written in the scientific principles (Clarity, brevity, chronological flow and attractiveness).

6. PLOS authors have the option to publish the peer review history of their article (what does this mean?). If published, this will include your full peer review and any attached files.

Reviewer #1: **Yes: **E. Fuller Torrey

Reviewer #2: **Yes: **SISAY ABEBE DEBELA

---

## [Author Response · Author response to Decision Letter 0]

5 Mar 2023

A point-by-point response for academic editor

Dear academic editor 

First, we would like to express our deepest gratitude for your constructive and valuable comments and suggestions to the best quality of our manuscript. As per your comments and suggestions, corrections have been made to the manuscript accordingly. 

Comment 1: “Please ensure that your manuscript meets PLOS ONE's style requirements, including those for file naming. The PLOS ONE style templates can be found athttps://journals.plos.org/plosone/s/file?id=wjVg/PLOSOne_formatting_sample_main_body.pdf andhttps://journals.plos.org/plosone/s/file?id=ba62/PLOSOne_formatting_sample_title_authors_affiliations.pdf”

Authors’ response: Thanks for your great comments. We have revised the manuscript as per the PLOS ONE's style requirements.

Comment 2: “In your Data Availability statement, you have not specified where the minimal data set underlying the results described in your manuscript can be found. PLOS defines a study's minimal data set as the underlying data used to reach the conclusions drawn in the manuscript and any additional data required to replicate the reported study findings in their entirety. All PLOS journals require that the minimal data set be made fully available. For more information about our data policy, please see http://journals.plos.org/plosone/s/data-availability”.

Authors’ response: According to your constructive comments, all data are fully available without restriction.

Comment 3: “Your ethics statement should only appear in the Methods section of your manuscript. If your ethics statement is written in any section besides the Methods, please delete it from any other section”.

Authors’ response: We appreciate your great concern. Some reptations were omitted. 

Comment 4: Please include a caption for figure 1.

Authors’ response: Thankfully, we have corrected it. Caption is included in figure 1.

Comment 5: “Please ensure that you refer to Figure 1 in your text as, if accepted, production will need this reference to link the reader to the figure”.

Authors’ response: Thankfully, we have corrected it.

Comment 6: “We note you have included a table to which you do not refer in the text of your manuscript. Please ensure that you refer to Table 3 in your text; if accepted, production will need this reference to link the reader to the Table”.

Authors’ response: Thankfully, we have corrected it. 

Comment 7: “Please include captions for your Supporting Information files at the end of your manuscript, and update any in-text citations to match accordingly. Please see our Supporting Information guidelines for more information: http://journals.plos.org/plosone/s/supporting-information”.

Authors’ response: Thankfully, we have corrected it. Captions are included for the supporting information file in the revised manuscript.

A point-by-point response for Reviewer 1

Dear Reviewer 

First of all, we would like to express our heartful thanks to the corrections, comments and suggestions you have forwarded for improving the quality of our manuscript. Dear reviewer, we have accordingly incorporated the whole things that you have raised and you can recheck the corrections made in the revised manuscript and the response letter attached.

Comment 1: The authors are to be congratulated for taking on this important subject. Shared decision-making between psychiatric patients and mental health workers is as important in Ethiopia as it is in England or America. The major impediment to achieving shared decision-making everywhere is the constraint of time; the less time per patient, the more difficult it is to achieve. Shared decision-making assumes that patients believe that there is something wrong and therefore need for decisions regarding treatment. If patients have an impaired awareness of their illness – anosognosia - shared decision-making is impossible to achieve other than agreeing to do nothing.

The main shortcoming of this is that it does not deal with this issue sufficiently. It simply says that “those patients who were seriously ill and had no insight were excluded.” How many were excluded? What percentage of the total sample?

Authors’ response: - By thanking you for your constructive comment, we have corrected it. Four patients were excluded due to they did not give response due to their active psychopathology. They were referred to the Emergency Psychiatry ward.

b. “Various levels of impaired insight occur most commonly in psychosis. Therefore if a mental health worker perceives or believes that a given patient has impaired insight, the mental health worker is less likely to attempt to achieve shared decision-making. If this is true that I would expect shared decision-making to be lower for patients with a diagnosis of schizophrenia, for example, and for those with a diagnosis of depression or anxiety disorders, for example. This could be easily tested with your existing data”

Authors, response: We really appreciate your suggestions to boost the quality of our manuscript. Based on your suggestion, the manuscript is revised. From two hundred six patients with mental illness, 39.3%, 27.2%, 18.9% and 7.8% had poor decision making involvement during their psychiatric outpatient follow-up among patients diagnosed with Schizophrenia, Major Depressive Disorder (MDD), Bipolar disorder (BPI disorder), and Generalized Anxiety Disorder (GAD) respectively (Fig 2).

Comment 2: “The paper could be improved grammatically by using a native English speaker. And there are a few typos such as ages listed in table 1 and the lack of publication information for reference number 16.

Authors’ response: We are very much thankful for your comment. We have corrected it accordingly. DOI of Reference number 16 is https://psycnet.apa.org/doi/10.2975/34.1.2010.7.13

A point-by-point response for Reviewer 2

Dear Reviewer 

First of all, we would like to express our heartfelt thanks to the corrections, comments and suggestions you have forwarded for improving the quality of our manuscript. Dear reviewer, we have accordingly incorporated the whole things that you have raised and you can recheck the corrections made in the revised manuscript and the response letter attached.

Comment 1: “The introduction is not clear enough. In this sense, you need to rewrite your introduction. I also think that there needs to be more clarity about the justification of the study and why you chose to use explanatory sequential mixed method. In general, the authors should revise their manuscript in a scientific way with correct language usage and resubmit their manuscript. Therefore, you should rewrite your introduction by focus on your topic taking into consideration the following points that to be raised. 1. What is it? The problem, 2.Magnitude and distribution?, 3. Possible factors?, 4. Severity and consequences? and 5. Gap statement?”

Authors’ response: We are very much thankful for your comment. We modified it based on your great suggestion.

Comment 2: “Keywords: Minor edition; make the first letters of all keywords uppercase.”

Authors’ response: We are so thankful for your insight. We have corrected it accordingly.

Comment 3: Methods

a. Study design?

Authors’ response: We are very much thankful for your comment. We have used Explanatory Sequential Mixed Method study. Since there were limited studies regarding shared decision making in mental health, its aim is to use qualitative approach to explain and support quantitative results. 

b. No needs to present sample size formula rather narrate how the sample size calculated. Remove the formula and definition of symbols or alphabet used in the sample size calculation.

Authors’ response: We are very much thankful for your comment; we modified it based on your great suggestion.

c. Figure 1 is the not figure in the scientific writing. Please, recreate figure 1 in attractive way by arranging the alignment of arrow and rectangles used.

Authors, response: We really appreciate your suggestions to boost the quality of our manuscript. Based on your suggestion, we have modified it. 

d. Where are study variables?

Operational definition: If you present operational definition for the outcome variable “Shared decision-making (level)” (What it mean low or high Shared decision making) in methodology part is more appropriate for scientific paper, and the way you measured the outcome variable (scoring, including categorization and how composite index or total score formed) under data collection with the variables measured. Thus, I recommend presenting the operational definition of Shared decision making level, and providing the method of measuring the Shared decision-making. Most importantly provide your justification why mean score was used as cut-off point with reference. Who recommend the cut-off point?

Authors’ response: With heartfelt thanks, study variables and operational definitions of each variable are added in the revised manuscript. Since it is a continuous variable the validator of the tool recommends to use the mean score (the reference is cited in the tool of the outcome variable https://doi.org/10.1016/j.pec.2009.09.034)

Comment 4: Result

a. Change Quantitative finding to socio-demographic characteristic of the respondents Authors’ response: 

Authors’ response: We are very much thankful for your comment. It was writing error and we have corrected it accordingly.

b. If you are presenting the exact number remove the words “about, nearly, majority…”, Because you are writing the exact number and the reader can evaluate by themselves.

Authors’ response: We are very much thankful for your insight. We have corrected it accordingly

c. Results presentation is not presented in the way readers can understand. Just a list of sentences without creativity. The authors should read previously published similar article and represent the results part clearly.

Authors, response: We really appreciate your suggestions to boost the quality of our manuscript. Based on your suggestion, we have modified it. The result is presented in tables, sentences and figures.

Comment 5: Discussion

The discussion part is not presented in the way guide the readers and clear way. In the first part of the discussion part, the authors should summarize the key findings which they are going to discuss. However, the authors just stated their discussion by comparing their finding with the previous findings. Even the way they are comparing with previous finding is not clear. Thus, the authors should read the previous study or any scientific writing books and try to summarize the key findings in the first paragraph of the discussion and then start results interpretation, and compare and contrast with previous studies. I strongly recommend for the authors should follow the previous study and try to present the discussion. In fact not only the discussion whole manuscript should be written in the scientific principles (Clarity, brevity, chronological flow and attractiveness).

Authors’ response: We really do appreciate your comments and all what you have mentioned is right and accepted. We have modified the discussion and the whole manuscript.

---

## [Decision Letter · Decision Letter 1]

21 Mar 2023

Level of shared decision making and Associated Factors among Patients with Mental Illness in Northwest Ethiopia: Explanatory sequential mixed method study

PONE-D-22-33947R1

Dear Dr. Abate,

We’re pleased to inform you that your manuscript has been judged scientifically suitable for publication and will be formally accepted for publication once it meets all outstanding technical requirements.

Kind regards,

Mesfin Gebrehiwot Damtew

Academic Editor

PLOS ONE

Additional Editor Comments (optional):

Reviewers' comments:

Reviewer's Responses to Questions

**Comments to the Author**

1. If the authors have adequately addressed your comments raised in a previous round of review and you feel that this manuscript is now acceptable for publication, you may indicate that here to bypass the “Comments to the Author” section, enter your conflict of interest statement in the “Confidential to Editor” section, and submit your "Accept" recommendation.

Reviewer #1: All comments have been addressed

Reviewer #2: All comments have been addressed

2. Is the manuscript technically sound, and do the data support the conclusions?

Reviewer #1: Yes

Reviewer #2: Yes

3. Has the statistical analysis been performed appropriately and rigorously? 

Reviewer #1: I Don't Know

Reviewer #2: Yes

4. Have the authors made all data underlying the findings in their manuscript fully available?

Reviewer #1: Yes

Reviewer #2: Yes

5. Is the manuscript presented in an intelligible fashion and written in standard English?

Reviewer #1: Yes

Reviewer #2: No

6. Review Comments to the Author

Reviewer #1: (No Response)

Reviewer #2: (No Response)

7. PLOS authors have the option to publish the peer review history of their article (what does this mean?). If published, this will include your full peer review and any attached files.

Reviewer #1: **Yes: **E. Fuller Torrey

Reviewer #2: **Yes: **Sisay Abebe Debela

---

## [Editor Report · Acceptance letter]

29 Mar 2023

PONE-D-22-33947R1 

Level of shared decision making and Associated Factors among Patients with Mental Illness in Northwest Ethiopia: Explanatory sequential mixed method study 

Dear Dr. Abate:

I'm pleased to inform you that your manuscript has been deemed suitable for publication in PLOS ONE. Congratulations! Your manuscript is now with our production department. 

Kind regards, 

on behalf of

Dr. Mesfin Gebrehiwot Damtew 

Academic Editor

PLOS ONE